# Frequency Shifts in Muscle Activation during Static Strength Elements on the Rings before and after an Eccentric Training Intervention in Male Gymnasts

**DOI:** 10.3390/jfmk7010028

**Published:** 2022-03-11

**Authors:** Beat Göpfert, Christoph Schärer, Lisa Tacchelli, Micah Gross, Fabian Lüthy, Klaus Hübner

**Affiliations:** 1Department Biomedical Engineering (DBE), University of Basel, 4001 Basel, Switzerland; 2Swiss Federal Institute of Sport Magglingen (SFISM), 2532 Magglingen, Switzerland; christoph.schaerer@baspo.admin.ch (C.S.); micah.gross@baspo.admin.ch (M.G.); fabian.luethy@baspo.admin.ch (F.L.); klaus.huebner@baspo.admin.ch (K.H.); 3Movement and Sport Science, Department of Medicine, University of Fribourg, 1700 Fribourg, Switzerland; lisa.tacchelli@bluewin.ch

**Keywords:** eccentric–isokinetic training, wavelet-transformed electromyogram, performance in gymnastics

## Abstract

During ring performance in men’s gymnastics, static strength elements require a high level of maximal muscular strength. The aim of the study was to analyze the effect of a four-week eccentric–isokinetic training intervention in the frequency spectra of the wavelet-transformed electromyogram (EMG) during the two static strength elements, the swallow and support scale, in different time intervals during the performance. The gymnasts performed an instrumented movement analysis on the rings, once before the intervention and twice after. For both elements, the results showed a lower congruence in the correlation of the frequency spectra between the first and the last 0.5 s interval than between the first and second 0.5 s intervals, which was indicated by a shift toward the predominant frequency around the wavelet with a center frequency of 62 Hz (Wavelet W10). Furthermore, in both elements, there was a significant increase in the congruence of the frequency spectra after the intervention between the first and second 0.5 s intervals, but not between the first and last ones. In conclusion, the EMG wavelet spectra presented changes corresponding to the performance gain with the eccentric training intervention, and showed the frequency shift toward a predominant frequency due to acute muscular fatigue.

## 1. Introduction

During ring performance in men’s gymnastics, static strength elements such as the swallow or the support scale (see Figure 1) require high levels of muscular strength, muscular control, and balance. The muscular requirements that are needed to overcome gravity, while holding a stable position for the required duration of two seconds are partly considered as an eccentric muscle contraction, which is highly demanding in terms of neuromuscular performance [1]. With time, the high muscle forces encountered during these elements inevitably induce muscle fatigue or complete exhaustion. As such, gymnastic coaches have employed specific eccentric training programs with a very slow eccentric velocity of 0.1 m/s to efficiently overload muscles, with the intention of improving ring-specific muscular strength [1]. The result of eccentric training during this study was seen to result in a significant improvement of the specific maximum strength [1]. Furthermore, the influence of eccentric training showed some changes in muscle properties, such as fiber composition [2], which were detectable in the frequency spectra of the electromyogram (EMG) [3,4]. In addition, the frequency spectra contain information about muscular activation patterns [5], fiber-type composition [6], and level of fatigue [7].

Although overall body position remains more or less static during hold elements, the unstable rings require constant adjustment of muscle activation, often in short intense bursts, in order to correct body position. This demand is in addition to the generally very high muscle forces needed to resist the weight and leverage of the body, which can lead to severe fatigue or exhaustion within a few seconds. Thus, whereas EMG indicators of muscle fatigue have been described previously for controlled isokinetic, isometric, or cyclic movements, hold elements on the rings represent a unique type of muscular performance, which has yet to be investigated. Furthermore, the effects of the abovementioned strength training intervention, aimed specifically at improving hold strength on the rings on EMG indicators of neuromuscular control, have not been explored.

Therefore, we aimed to describe the changes of the wavelet-transformed muscle intensity pattern and frequency spectra of eight upper body muscles, while maintaining the static strength elements of the swallow and support scale on rings. Additionally, we investigated changes in EMG indicators of neuromuscular control before and after a four-week specific eccentric strength training intervention with a gymnastic-specific eccentric–isokinetic (0.1 m/s) cluster training on a computer-controlled training device [1].

We hypothesized that muscular failure during hold elements on the rings would correspond to shifts in the relative intensities of frequency bands of wavelet-transformed EMG signals, and that training would prevent or limit such shifts for the muscle forces needed to perform the two static strength elements.

## 2. Materials and Methods

### 2.1. Subjects

Eight international or national top-level gymnasts (ethnic group: Caucasian; age: 21.47 ± 1.96 years; height: 169.84 ± 5.47 cm, and weight: 69.4 ± 7.0 kg) volunteered to participate in this study. All athletes were members of their national team, trained approximately 24 h per week, and were free of any injury at the time of the intervention. They were informed of the benefits and risks of the investigation prior to signing an informed consent to participate in the study. The study was approved by the Ethics Committee of Bern (Project-ID: 2017-01891) and conducted in accordance with the current version of the Declaration of Helsinki, the ICH-GCP ISO EN 14,155, and all national legal and regulatory requirements.

### 2.2. Data Collection and Processing

Data collection occurred during late winter at three time points within the athletes’ periodized basic training. The measurements took place one week before (T1) and at one (T2) and three (T3) weeks after the four-week gymnastic-specific eccentric–isokinetic cluster training intervention (0.1 m/s), which is described in detail elsewhere [1]. Each data collection session took place at the training facility of the gymnastics national team and began with an individual warmup. Thereafter, gymnasts performed the elements of the swallow and support scale on the rings (see Figure 1) for five seconds under the same conditions for all three measurement sessions (for details, see [1]).

Between trials, gymnasts passively rested for at least five minutes. A given athlete’s resistance was kept constant for all time points. Although in many cases after the training intervention (at T2 and T3) athletes were able to perform the elements with this resistance for longer than five seconds, only the first five seconds were analyzed for this study.

Between warmup and performance trials, surface EMG electrodes (Type F3010, FIAB Spa, Vicchio, Italy) were placed on the Mm. biceps brachii (BB), pectoralis major (PM), deltoideus anterior (DA) and posterior (DP), seratus anterior (SA), infraspinatus (IS), trapezius transversalis (TT), and trapezius ascendens (TA). Each of the electrodes was connected to a wireless EMG transmitter (Myon, M320, Myon AG, Switzerland). The skin preparation and electrode placement were performed according to the SENIAM standards [8,9]. Additionally, reflective markers (Vicon Motion Systems Ltd., Oxford, UK) were placed on the body according to the manufacturer’s full-body plug-in gait model. During the performance trials, EMG data were sampled at a frequency of 2400 Hz, while 3-D motion of the reflective markers was captured by eight infrared cameras (Vicon Vantage 5, Vicon Motion Systems Ltd., Oxford, UK) at 120 Hz. The cameras and the EMG receiver were connected to a desktop computer running Vicon Nexus (version 2.5) software, which synchronously recorded the EMG and motion-capture data from each trial.

Motion-capture data were used to control the position of the hold elements. The trials were valid if the angular deviation of the elements performed was smaller than 10° from the starting position, in accordance with the requirements for recognition of strength elements in the International Gymnastics Federation’s code of points [10]. The raw EMG data were transformed using a wavelet transformation [11] with 23 wavelets (W1 to W23) with the following center frequencies (fc): 1.3, 3.6, 7.1, 11.6, 17.3, 24.1, 31.9, 40.9, 50.9, 62.0, 74.1, 87.4, 101.6, 117.0, 133.3, 150.8, 169.3, 188.8, 209.4, 231.0, 253.6, 277.3, and 302.0 Hz. After the wavelet transformation, signals were time-normalized to 1000 values per trial, to allow comparison despite small (<0.5 s) differences in the performance duration. Initially, multi-muscle intensity patterns (MMIP) from wavelets 3 (fc 7.1 Hz) through 23 (fc 302.0 Hz) were generated and visually inspected for signal quality (e.g., no signal due to insufficient electrode–skin contact caused by sweating or wear). Thereafter, trials were split into 10 equal segments of 100 values. The frequency spectra of each 10% interval and muscle were determined, and the intensity of wavelets 3–23 was normalized to the frequency spectrum of that interval, thus yielding a relative intensity for each wavelet. This allowed a comparison between the relative intensity distributions of wavelets 3–23 at different time intervals for each muscle of a subject. The similarity of the frequency spectra was analyzed using Pearson’s correlation coefficient (*r*) on the relative intensities of wavelets 3–23 of each subject’s muscle at two time intervals, where *r* = 1 would indicate perfect correlation (no shifts) in the relative intensities of wavelets between time intervals. Specifically, congruence in relative wavelet intensities was assessed between the first (I1, 0–10% of trials duration) and second (I2, 10–20%) time intervals, and between I1 and the last (I10, 90–100%) time interval. In this manner, incongruences (i.e., shifts) in relative wavelet intensities occurring from the beginning to the end of a trial (r_end_, from I1 vs. I10) could be compared with shifts occurring between the first second of the trial intervals (r_contro_l, from I1 vs. I2), where r_control_ served as a surrogate control value, for which maximal congruence (i.e., a high *r*-value) was expected. To assess the first research question (acute effects), a group comparison with all muscles pooled (*n* = 8 × 8 = 64, i.e., one *r*-value per muscle, 8 *r*-values per athlete) between r_end_ and r_control_ was made using the Mann–Whitney *U*-Test (α = 0.05). Thus, *p*-values < 0.05 indicate, for the subject cohort as a whole, significant differences in wavelet relative intensities at the end compared to the beginning of the hold element, namely, a significant shift in wavelet intensity distribution over the course of a trial. Moreover, for the second research question (chronic effects), multiple comparisons were made among r_control_ from time points T1, T2, and T3, as well as among r_end_ from these three time points. Again, group data with all muscles pooled (*n*= 8 × 8 = 64, i.e., one *r*-value per muscle, 8 *r*-values per athlete) were compared using the Mann–Whitney *U*-Test (α = 0.05).

## 3. Results

### 3.1. Visual Inspection of MMIP

The MMIPs showed muscle- and subject-specific activation patterns. Some of the variations observed during the visual quality inspection were caused by the slightly different (generally improved) body position during the performance of the static strength elements from one time point to the next. Additionally, the intensity patterns varied between the two elements, as they demand a slightly different muscular activation. For example, for the athlete shown in Figure 2, the bandwidths of activation for the Mm. biceps brachii and seratus anterior were different between the two elements. For the swallow, the muscular activity pattern showed a higher intensity over a broader frequency range compared to the support scale. This is most likely related to the difference in arm position between the two elements. Furthermore, for the swallow, Mm. trapezius ascendens, infraspinatus, and trapezius transversalis showed visibly lower intensity value at the beginning, which then increased to between 40 and 50% of the trial duration in the narrower frequency range; moreover, after 60% of the trial duration, these three muscles displayed synchronized local total intensity peaks, so-called synchronized muscular events (Figure 2, dotted squares) [4]. In contrast, for the support scale, only M. trapezius transversalis showed greater increasing activity, whereas Mm. trapezius ascendens and infraspinatus exhibited similar high activity throughout the entire trial, as for the swallow.

Furthermore, in several athletes, independent of the element performed, MMIPs revealed differences in activation between muscles in relation to the bandwidth of the active wavelets. In general, the frequency spectrum changed from a broad to a narrow one, with wavelet W10 with a center frequency fc 62 Hz being the most dominant. Additionally, muscular events predominated towards the end of trials. For example, M. pectoralis major (Figure 3) was active in a broader frequency range than Mm. trapezius ascendens and deltoideus anterior, which might be related to their muscular structures (e.g., their fiber characteristics) and loading conditions. Moreover, some muscles showed a distinct narrowing of their frequency range over the course of the trial (Figure 3, muscle deltoideus posterior, pectoralis major, red lines). Shortly prior to five seconds (i.e., at the point of exhaustion for T1), most athletes displayed synchronous activity and simultaneous muscle events for several muscles (Figure 3, Arrow a).

### 3.2. Correlations

For both the swallow (r_control_ = 0.93 ± 0.06, r_end_ = 0.85 ± 0.10) and support scale (r_control_ = 0.90 ± 0.08, r_end_ = 0.82 ± 0.12), congruence of the wavelet relative intensity distribution decreased significantly (*p* < 0.01) over the course of all trials. Comparing time points, there was a significant increase (*p* < 0.01) in r_control_ for swallow between T1 (0.92 ± 0.05) and T2 (0.94 ± 0.08), and T1 to T3 (0.95 ± 0.04) but no change in r_end_ (0.847 ± 0.10 at T1, 0.846 ± 0.11 at T2, and 0.85 ± 0.09 at T3). Similarly, for the support scale, a significant increase (*p* < 0.01) in r_control_ was observed between T1 (0.87 ± 0.10) to T2 (0.91 ± 0.06) and T1 to T3 (0.91 ± 0.07). Furthermore, there was also no significant increase in r_end_ between T1 (0.78 ± 0.18), T2 (0.84 ± 0.09), and T3 (0.83 ± 0.11).

A look at individual muscles revealed the largest chronic effect for m. deltoideus anterior. For this muscle, r_end_ increased (*p* < 0.01) between T1 and T2 for both the swallow (from 0.78 ± 0.13 to 0.85 ± 0.09) and support scales (from 0.72 ± 0.21 to 0.82 ± 0.11) (Figure 4). Changes between T2 and T3, and in other muscles, displayed no clear tendencies.

## 4. Discussion

This study investigated acute and chronic changes in the frequency spectra of wavelet-transformed EMG signals obtained from gymnasts performing hold elements on the rings, in order to describe the effects of fatigue and strength-training adaptations, respectively, on muscle activation patterns. Regarding acute fatigue, we showed poorer congruence between the last and the first 0.5 s intervals than between the second and first 0.5 s intervals, indicating a clear shift in the predominant frequency of all measured muscles around wavelet W10 with a center frequency fc 62 Hz over the course of the trial. Specifically, the results showed a shift toward a lower predominant frequency as hypothesized, which has been described previously as a typical sign of muscular fatigue [7,13,14]. A second sign of acute fatigue occurred toward the end of the five-second trial, which is muscular activation within a narrow frequency bandwidth in most muscles. This was visible in the MMIP as highly intense muscular events within a narrow band centered on wavelet 10 with a center frequency of 62 Hz. This effect has been described to indicate the synchronization of the motor units [7,11,15].

The synchronization bandwidth in this study was similar to that measured during an isometric contraction of m. abductor pollicis brevis by Barandun et al. [11] but different from that observed for gait [7] or running [15]. Thus, the fatigue-related synchronization band is likely to be related to the task performed, muscle properties such as fiber-type composition [6], and the ability to synchronize the muscle [4].

Additionally, some athletes displayed simultaneous muscular evens in the MMIP (Figure 3), which indicate a synchronized muscular activation over several muscles or muscular groups (e.g., TA, IS, and TT, or DL and DA). A similar effect has been described by Stirling et al. [16] for M. gastrocnemius medialis during running and by Huber et al. [17] for the different parts of the quadriceps muscle during walking. Although the two static strength elements are non-cyclic high-force movements, it would be very difficult in the present study to distinguish clearly between neuromuscular control phenomena, such as Piper rhythm [18] and muscular fatigue.

The four-week eccentric training intervention led to increased muscular strength and ring performance as measured by Schärer et al. [1]. This was accompanied by improved time-to-exhaustion while performing hold elements with the pretraining maximal resistance. This increased stamina in the pretraining maximal resistance could be expected to confirm better-maintained EMG wavelet relative-intensity distribution over the course of the first five seconds. We investigated this hypothesis by comparing the degrees of congruence between wavelet relative intensities at the beginning and the end of the five-second trials (r_end_) at pre- and post-training time points. There was an increase after the training intervention in r_control_ for both elements (which served as a surrogate control parameter for comparison with r_end_ at a single given time point), which could indicate more stable neuromuscular control during the first second of a trial, and an increase in r_end_ was only observed for the support scale (between T1 and T2), whereas none was observed for the swallow scale. As the support scale could be considered to be the co-ordinatively more demanding of the two elements (as the center of mass is higher and the arm positions require more balance than in the swallow element), it appears that the EMG analyses employed in this study were most able to detect training effects under conditions of higher muscular strength demands [1]. Interestingly, although the training intervention was very demanding and fatiguing, there were no further changes in the neuromuscular control parameters employed in this study between T2 and T3.

The study has a small subject group, because of the availability of top athletes, and therefore has limited power. Nevertheless, it shows that there are different levels of learning in the static force elements. In addition to the required maximum force, learning the ability of neuromuscular activation and control of the muscles during training [19] is likely to play an important role in how a static force element can be executed economically and correctly.

## 5. Conclusions

This study showed that changes in EMG wavelet intensity distribution, specifically to intensity converged into a narrower frequency band, corresponded with acute muscular fatigue in male gymnasts during hold elements on the rings. These holding elements are a form of exhaustive isometric muscle action with a high stabilization and balance component. Regarding training-induced adaptations, improved performance for both hold elements, maximal resistance, and hold time at the initial maximal resistance [20] only partly corresponded with the EMG-based indicators of neuromuscular control and fatigue resistance used in this study. Specifically, the hypothesized better conservation of relative frequency distribution over the course of a five-second trial was observed for the support scale but not for the swallow.

## Figures and Tables

**Figure 1 jfmk-07-00028-f001:**
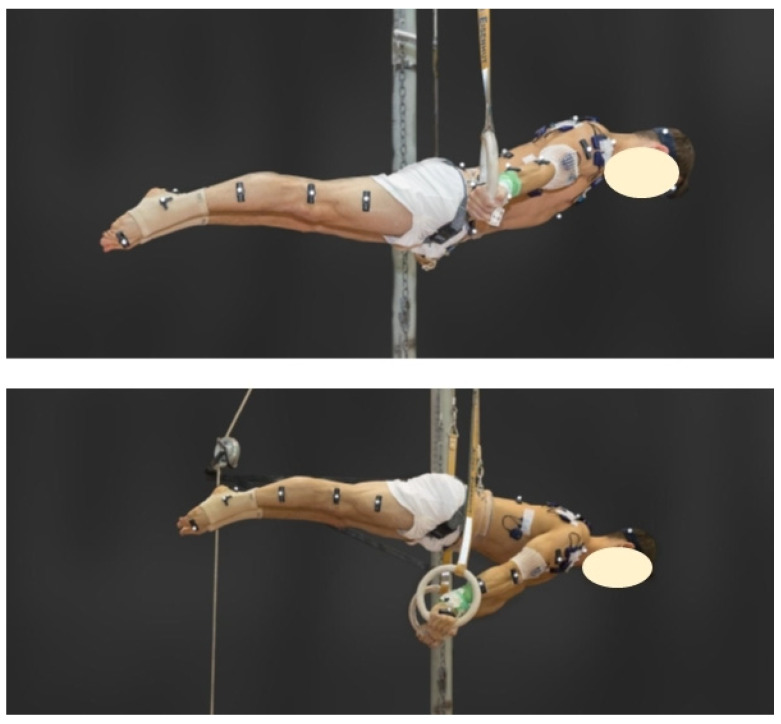
Hold positions for the elements swallow (**top**) and support scale (**bottom**).

**Figure 2 jfmk-07-00028-f002:**
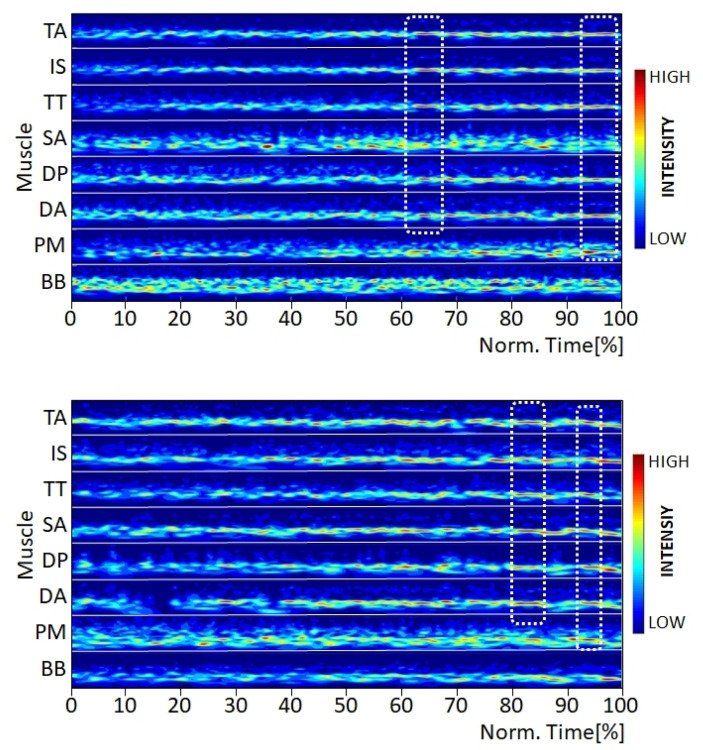
Multi-muscle intensity pattern (MMIP) of the wavelet-transformed EMG signal for the elements swallow (**top**) and support scale (**bottom**) from the same subject at T1. Note: Each muscle is normalized to its maximal amplitude [12].

**Figure 3 jfmk-07-00028-f003:**
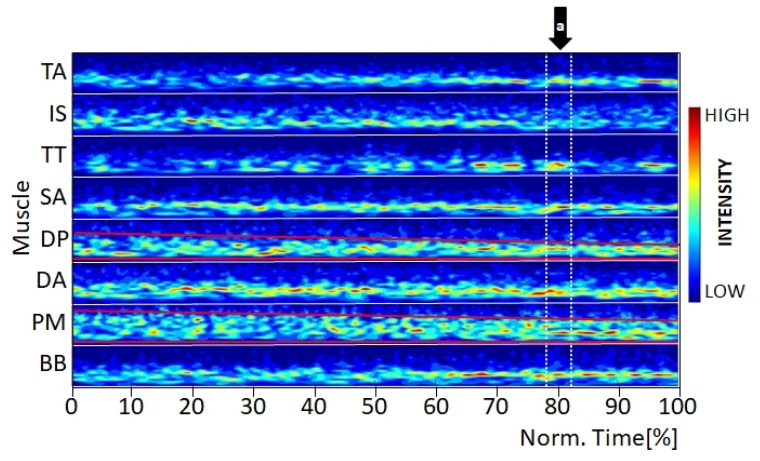
Multi-muscle intensity pattern (MMIP) of the wavelet-transformed EMG signal for the element support scale, displaying a narrowing of the frequency bands of the muscles deltoideus anterior (DL) and pectoralis major (PM). Arrow a: Simultaneous muscle events for seven of the eight measured muscles. Note: Each muscle is normalized to its maximal amplitude [12].

**Figure 4 jfmk-07-00028-f004:**
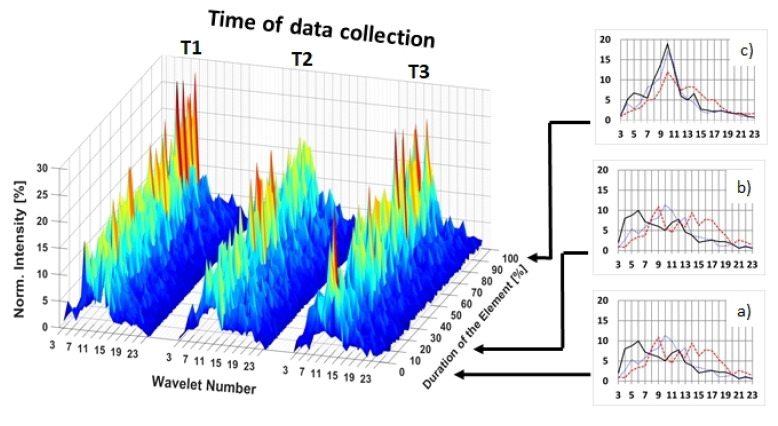
**Left**: Normalized 3D-intensity pattern for the element support scale and the muscle deltoideus anterior (DA) at time points T1, T2 and T3. **Right**: the average frequency spectra of 10%-time-intervals I1 (**a**), I2 (**b**), and I10 (**c**).

## Data Availability

The data that support the findings of this study are available from the corresponding author, B.G., upon reasonable request.

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
