# Peer review of "Frequency Shifts in Muscle Activation during Static Strength Elements on the Rings before and after an Eccentric Training Intervention in Male Gymnasts"

_jfmk, 2022, doi:10.3390/jfmk7010028_

Round 1

Reviewer 1 Report

The manuscript represents an original and significant contribution that  analyze, ring performance in men’s gymnastics, the effect of a four-week eccentric isokinetic training intervention in the frequency spectra of the wavelet transformed electromyogram (EMG) during the two static strength elements, the swallow and support scale, in different time intervals during the performance.
The general organization of the review is solid, references are comprehensive and updated, the experimental evidences are well interpreted and, finally, the functional considerations are original.

Author Response

Dear Reviewer,

Thank you very much for your work and time.
We added the comments of Reviewer 2 in the article.
-  ethnic group
-  sample size and practical implications:

"Although the study has a small subject group, because of the availability of top athletes and therefore has a limited power. Nevertheless, it showed that there are different levels of learning in the static force elements. In addition to the required maximum force, learning the ability of neuro-muscular activation and control of the muscles during training [19] is likely to play an important role in how a static force element can be executed economically and correctly."

19.    Gabriel, D. A., Kamen, G., & Frost, G. (2006). Neural adaptations to resistive exercise. Sports Medicine, 36(2), 133-149.

Reviewer 2 Report

This study deals with a very interesting topic, adding important information on acute and chronic changes in the frequency spectra of wavelet-transformed EMG signals in male gymnasts performing hold elements on the rings, and describe the effects of fatigue and strength-training adaptations on muscle activation patterns.

The design of the study is well structured, the methodology and statistical analyses are correct.

In addition, the results and conclusions are clear and exhaustive.

I have only some minor revisions to request.

Subjects (line 67): please add the ethnic group of the participants.

Discussion

Even if the study regards high-level athletes, I would insert the low sample size as a limit of the study.

Furthermore, I would reinforce the practical implications that this study entails.

Author Response

Dear Reviewer,

Thank you for your work and your comments.

We added the ethnic group: Caucasian in line 67

We added a section in the discussion noting the small subject number and the importance of learning neuo-muscular control during training for execution of the static force element.

"Although the study has a small subject group, because of the availability of top athletes and therefore has a limited power. Nevertheless, it showed that there are different levels of learning in the static force elements. In addition to the required maximum force, learning the ability of neuro-muscular activation and control of the muscles during training [19] is likely to play an important role in how a static force element can be executed economically and correctly."

19.    Gabriel, D. A., Kamen, G., & Frost, G. (2006). Neural adaptations to resistive exercise. Sports Medicine, 36(2), 133-149.

Further the English Editing Services of MDPI was used.